# Aquaculture at the crossroads of global warming and antimicrobial resistance

Miriam Reverter [1,2✉], Samira Sarter [1,3], Domenico Caruso[1], Jean-Christophe Avarre [1], Marine Combe[1], Elodie Pepey[1,3], Laurent Pouyaud[1], Sarahi Vega-Heredía [1], Hugues de Verdal [1,3] & Rodolphe E. Gozlan [1✉]

In many developing countries, aquaculture is key to ensuring food security for millions of people. It is thus important to measure the full implications of environmental changes on the sustainability of aquaculture. We conduct a double meta-analysis (460 articles) to explore how global warming and antimicrobial resistance (AMR) impact aquaculture. We calculate a Multi-Antibiotic Resistance index (MAR) of aquaculture-related bacteria (11,274 isolates) for 40 countries, of which mostly low- and middle-income countries present high AMR levels. Here we show that aquaculture MAR indices correlate with MAR indices from human clinical bacteria, temperature and countries' climate vulnerability. We also find that infected aquatic animals present higher mortalities at warmer temperatures. Countries most vulnerable to climate change will probably face the highest AMR risks, impacting human health beyond the aquaculture sector, highlighting the need for urgent action. Sustainable solutions to minimise antibiotic use and increase system resilience are therefore needed.

[1] ISEM, Univ Montpellier, CNRS, EPHE, IRD, Montpellier, France. [2] Institut für Chemie und Biologie des Meeres (ICBM), Carl von Ossietzky Universität Oldenburg, Wilhelmshaven, Germany. [3] CIRAD, UMR ISEM, F-34398 Montpellier, France. ✉email: mirireverter@gmail.com; rudy.gozlan@ird.fr

A key challenge for the years to come is feeding a rapidly growing human population while lowering the impact of food production on the environment[1]. This is particularly true for low- and middle-income countries (LMICs) where the demand for animal protein is likely to rise[2] and where existing environmental changes (e.g. droughts, floods, extensive wildfires) have in recent years led to major food crises[3,4]. As such, food security is central to the 2030 UN Agenda of Sustainable Development Goals, which aim to end poverty and to protect the planet from degradation[1]. To achieve these goals, food production not only needs to be increased, but most of all, good husbandry practice must follow to reduce its negative impacts on the environment. Currently, the typical response to this increased food demand is the intensification of production, underpinning environmental and health hazards such as increased water needs or overuse of antimicrobials[5–7].

Several studies have suggested that shifting human diet towards increased consumption of fish and seafood could be a solution to the need for protein that would sustain human and environmental health[8–10]. In fact, fish and seafood consumption is forecast to increase by 27% on the horizon of 2030, mostly sustained by the aquaculture sector, which is expected to grow by 62% during the same period[11]. The aquaculture industry contributes significantly to the livelihood of many households, with over 100 million people estimated to rely on aquaculture for their living[12]. As such, aquaculture plays a significant role in food security and poverty alleviation[13,14]. However, fish farming relies heavily on the use of antibiotics to combat infectious diseases that threaten production, with emerging infectious diseases (EIDs) forecast to increase with warmer temperatures[15–19]. For example, outbreaks of edwardsiellosis, streptococcosis and acute hepatopancreatic necrosis disease are often observed when temperature rises[20–22]. In this context, antimicrobial use is expected to rise in coming years, especially in LMICs, with the shift towards more intensified production systems to meet economic requirements and the demand for animal products[6,7]. However, the combined use of antimicrobial drugs in aquaculture and land-derived contamination into watercourses, contribute to the selection, emergence and spread of drug-resistant pathogens, posing an important public health threat[6,7,23].

Antimicrobial resistant (AMR) bacteria cause over 35,000 human deaths annually in the USA, 33,000 in the European Economic Area, 58,000 in India and probably more in SE Asia and these numbers are expected to rise[24,25] due to rapid socioeconomic development and population growth. Although the precise quantities of antimicrobials used in aquaculture are mostly unknown (especially in LMICs), antibiotic residues and AMR bacteria are often found in aquaculture environments[26–28]. Since aquatic environments are effective reservoirs of AMR bacteria from different sources[29] (e.g. human waste water, hospital effluents and animal and plant agricultural run-off), the direct contribution of aquaculture to this pool of AMR remains extremely hard to untangle. The aquaculture sector contributes to the AMR reservoir mainly by administering therapeutic and prophylactic antimicrobial treatments to animals but also to a minor extent with the use of non-antibiotic chemicals (e.g. disinfectants), which has been shown to increase AMR[30,31]. The presence of AMR in aquaculture production systems may not only pose a direct threat to human health, but could also impact production itself by lowering drug efficacy[23,32], decreasing the animal's immune system as seen in rats[33] and selecting more virulent strains (i.e. faster growth and higher transmission rates of pathogens)[34]. Recent research has shown that antimicrobial use might not be the only factor behind selection and emergence of AMR and warmer temperatures have been associated with higher AMR rates in terrestrial bacteria, establishing a sombre prospect in light of global climate warming[35].

In this context, we investigate the complex interplay between global warming and the occurrence of AMR in aquaculture. We first perform a meta-analysis to study the temperature effect on the mortality of aquatic animals infected with pathogenic bacteria commonly found in aquaculture and we observe most infected cultivated aquatic animals present higher mortalities at warmer temperatures. Then, we conduct a systematic review on the abundance of AMR bacteria found in aquaculture environments and calculate the multi-antibiotic resistance (MAR) index for 40 countries, as the ratio between the number of resistant bacterial isolates (i.e. strains or species) and the total number of combinations tested (number of antibiotics * number of isolates tested). MAR indices from aquaculture-related bacteria are further correlated to environmental and socioeconomic indicators to map countries or regions that are most at risk of AMR increase. Our results show that most countries present high MAR indices of aquaculture-related bacteria and that these were related to MAR indices from human clinical bacteria, temperature and climate vulnerability. These results suggest countries most vulnerable to climate change will probably face the highest AMR risks, impacting human health beyond the aquaculture sector.

## Results

**Effects of temperature on aquatic animal mortality.** After screening the literature, we extracted data from a total of 273 studies to test the influence of temperature on the mortality of cultured aquatic animals experimentally infected with major bacterial pathogens: *Aeromonas spp.*, *Edwardsiella spp.*, *Flavobacterium spp.*, *Streptococccus spp.*, *Lactococcus spp.*, *Vibrio spp.* and *Yersinia spp.* After controlling for effects of study ID and pathogen and host taxonomy when necessary, linear mixed models showed that an increase in temperature is associated with an increase in mortality rates in most infected aquatic organisms (Table 1, Supplementary Tables 1–15 and Fig. 1). Only mortalities

**Table 1 Parameters from the selected linear mixed models (LMM) to test the relationship between aquatic animal mortality and temperature (*T*) under bacterial infections.**

| Data subset | Model selected | Adj. $R^2$ | Parameter | Estimate | SE | 95% CI | |
|---|---|---|---|---|---|---|---|
| | | | | | | Lower | Upper |
| Bacterial infections in warm water | Mortality ~ T + mode of infection*log (dose) + (1 \| Reference) | 0.615 | T | 3.48 | 0.33 | 2.82 | 4.12 |
| | | | Mode of infection: injection | −38.67 | 16.86 | −72.03 | −5.52 |
| | | | Log(dose) | −2.07 | 0.95 | −3.93 | −0.20 |
| | | | Mode of infection: injection * log(dose) | 3.16 | 1.16 | 0.88 | 5.46 |
| Bacterial infections in temperate water | Mortality ~ T + mode of infection + (1 \| pathogen species) | 0.460 | T | 4.93 | 0.53 | 3.87 | 6.00 |
| | | | Mode of infection: injection | 16.67 | 5.77 | 4.19 | 29.65 |

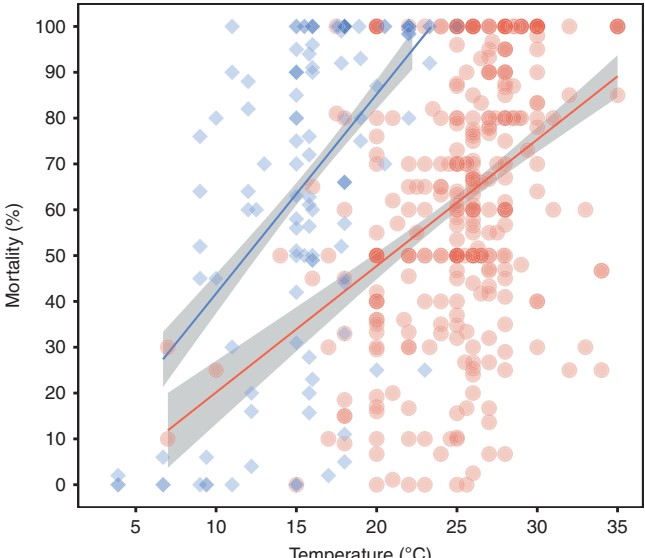

**Fig. 1 Predicted changes in mortality (%) of reared aquatic animals infected by bacterial diseases in response to temperature (°C).** Bacterial pathogens: *Aeromonas spp.*, *Edwardsiella spp.*, *F. columnare*, *Lactococcus spp.*, *Streptococcus spp.*, *Vibrio spp.*, and *Yersinia spp*. Red indicates tropical and subtropical host species ($n = 329$), blue indicates temperate host species ($n = 129$). Dots represent the raw data and the lines the linear mixed model predictions with SE.

of temperate fish infected with *Edwardsiella* spp. and *Y. ruckeri*, and warm water fish infected with *F. columnare* did not increase significantly with temperature (Supplementary Tables 3, 10, 13, 15 and Supplementary Fig. 1a–h). Models predicted that a temperature increase of 1 °C in warm-water and temperate organisms infected with bacteria could lead to increases of mortality of 2.82–4.12% and 3.87–6.00% respectively (95% of confidence) (Table 1, Fig. 1). Mode of infection and infecting dose were often important predictors of mortality outcome and were included in the models accordingly (Table 1 and Supplementary Table 15).

**AMR from aquaculture-related bacteria**. Antimicrobial resistance of 11,274 different bacterial isolates from aquatic reared animals and the aquaculture environment were obtained from literature (total of 130,426 antimicrobial resistance patterns to individual antibiotics). It is however important to highlight that, despite our best efforts in gathering a global database, the calculated aquaculture-derived MAR indices might be limited by the uneven report of antimicrobial resistances between different countries. These data were used to calculate the aquaculture MAR index for 40 countries, which accounted for 93% of global animal aquaculture production. Twenty-eight countries out of the 40 studied displayed MAR indices higher than 0.2, a threshold considered to be an indication of high-risk antibiotic contamination[36]. The mean global MAR index of aquaculture-related bacteria was 0.25 (SE = 0.01). Zambia (0.56) followed by Mexico (0.55) and Tunisia (0.53) were the countries with the highest MAR indices, whilst Canada (0.02), France (0.03) and USA (0.08) displayed the lowest (Fig. 2).

**Association between AMR and several indicators**. Antimicrobial resistance is thought to be a direct result of antimicrobial drug use. Since antibiotic use in aquaculture is neither harmonised nor consistently reported, it was not possible to further explore this relationship. However, the correlation between the MAR indices

obtained at country level for aquaculture-related bacteria and 20 environmental, health and socioeconomic indicators that could affect the emergence or spread of antibiotic resistance in the aquatic environment was tested. A strong positive correlation was found between human clinical MAR (MAR calculated from patient isolates[37]) and aquaculture-derived MAR indices (Table 2, Fig. 3a), but no correlation was found between aquaculture MAR indices and the use of clinical antibiotics (antibiotics sold in retail and hospital pharmacies for human consumption[38]) (Table 2). We also observed a negative relationship between gross domestic product (GDP) per capita, Human Development Index (HDI) and the Environmental Performance Index (EPI) with aquaculture-derived MAR indices ($r_{GDP} = -0.30$, $r_{HDI} = -0.30$, $r_{EPI} = -0.29$, $P = 0.06$). HSBC climate vulnerability index (CVI), an index that combines climatic and socioeconomic information to estimate countries' vulnerability to climate change (lower scores imply higher vulnerability)[39], was negatively correlated with aquaculture MAR indices ($r = -0.39$, $P = 0.02$, Table 2 and Fig. 3b). Correlations between the aquaculture MAR indices and the CVI showed that this association was underpinned by the physical impacts score (calculated from temperature levels, water availability and frequency of extreme weather events) as well as a country's ability to respond to climate change (Table 2). In addition, the average annual temperature (obtained from sites from which the AMR data were documented) positively correlated with the countries aquaculture MAR indices ($r = 0.37$, $P = 0.01$, Fig. 3c). None of the other indicators used (population density, animal production, animal trade, aquaculture production, reported pesticide use, % of undernourishment, % of people with access to at least basic sanitation services, % of people with access to at least basic water services) displayed a significant correlation (Table 2). Most of the LMICs (e.g. Vietnam, India, Pakistan, Bangladesh) simultaneously displayed the highest levels of human clinical and aquaculture-derived MAR and were the ones exposed to the highest climatic vulnerability and temperature rises, suggesting LMICs were regions most at risk for the combined action of global warming and AMR occurrence (Fig. 3). We further studied pairwise correlations between all simple (i.e. non-composite) variables to explore the relationships between them and the presence of underlying co-founding factors. The resulting correlation network shows that temperature, clinical MAR and GDP per capita were highly correlated between them as well as to several other socioeconomic and environmental variables (Fig. 4). In order to estimate the separate contribution of the simple variables (temperature, clinical Mar and GDP per capita) to aquaculture-related MAR and the effects of their interactions, a multiple regression model was applied using the LMG relative importance method to calculate the relative importance of each of the variables. After selection of the model explaining the highest variance (Supplementary Table 16), we observe that clinical MAR and temperature explain 17.9% and 9.1% of the aquaculture MAR, respectively, suggesting a direct involvement of these two variables to aquaculture-related antimicrobial resistance (Supplementary Fig. 2). The interaction between temperature and GDP per capita explained 9.7% of the MAR variability, whereas the GDP per capita only contributed to 3%, suggesting a complex interplay between different socioeconomic and environmental variables (Supplementary Fig. 2).

## Discussion

Our results show that aquaculture environments in most countries present high levels of AMR. We found a strong correlation between MAR indices from aquaculture and MAR indices from human clinical bacteria, suggesting that different activities (human, livestock and aquaculture antimicrobial consumption)

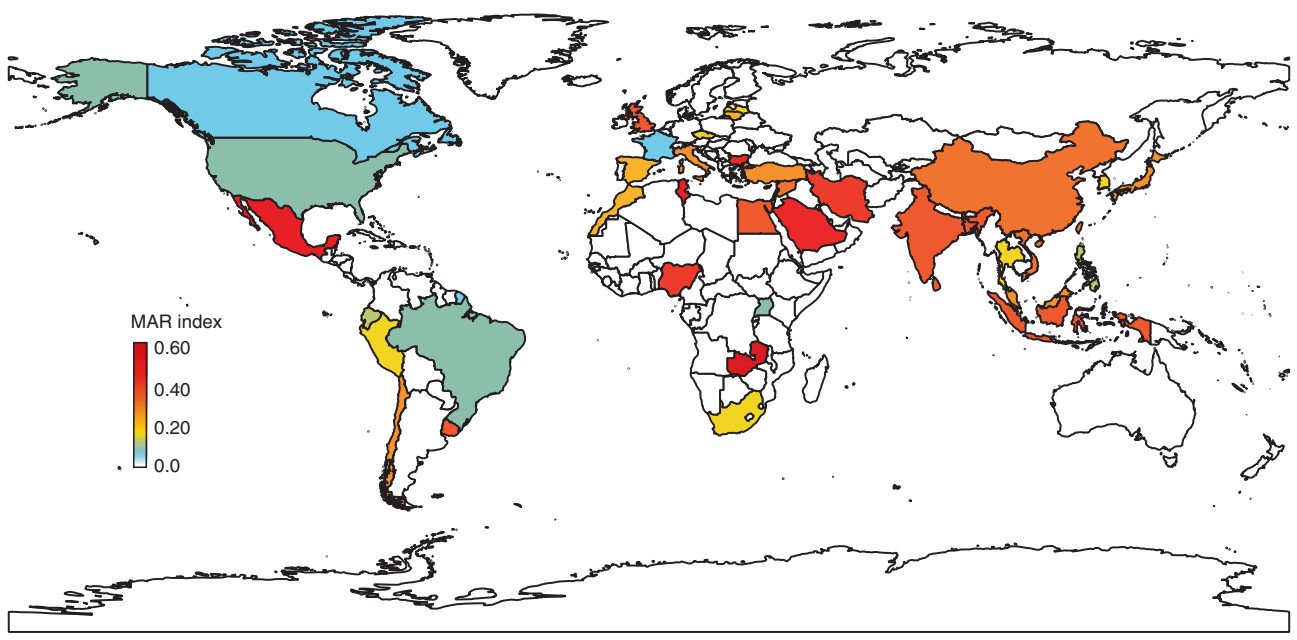

**Fig. 2 Global multi-antibiotic resistance (MAR) index calculated from aquaculture-derived bacteria.** No MAR index was calculated for countries in white due to data deficiency.

**Table 2 Pearson correlation (two-sided test, coefficient and 95% confidence intervals) between multi-antibiotic resistance (MAR) calculated indices from aquaculture-related bacteria and 20 environmental, health and socio-economic indicators.**

| Indicators | Pearson r | P-value | 95% CI | |
|---|---|---|---|---|
| | | | Lower | Upper |
| Climate vulnerability index | −0.39 | 0.02* | −0.65 | −0.05 |
| Physical impacts | −0.48 | 0.005* | −0.71 | −0.16 |
| Extreme events | 0.05 | 0.7 | −0.30 | 0.40 |
| Energy transition | −0.2 | 0.2 | −0.52 | 0.16 |
| Response to climate change | −0.42 | 0.02* | −0.67 | −0.08 |
| Temperature | 0.37 | 0.01* | 0.06 | 0.61 |
| EPI | −0.29 | 0.07 | −0.56 | 0.03 |
| GDP capita | −0.30 | 0.06 | −0.56 | 0.02 |
| Undernourishment | 0.16 | 0.33 | −0.16 | 0.45 |
| HDI | −0.30 | 0.05 | −0.57 | 0.01 |
| Pesticide use | 0.003 | 0.98 | −0.32 | 0.32 |
| Aquaculture production | 0.03 | 0.87 | −0.29 | 0.34 |
| Animal production | 0.07 | 0.67 | −0.25 | 0.37 |
| Animal trade | 0.12 | 0.47 | −0.41 | 0.20 |
| Livestock density | −0.07 | 0.97 | −0.32 | 0.31 |
| Total use of clinical antibiotics | 0.01 | 0.85 | −0.33 | 0.35 |
| MAR clinical | 0.58 | 0.001* | 0.27 | 0.78 |
| Basic sanitation services | −0.14 | 0.40 | −0.43 | 0.18 |
| Basic water services | −0.04 | 0.80 | −0.34 | 0.27 |
| Population density | 0.09 | 0.59 | −0.23 | 0.39 |

*Indicates statistical significance (P-value < 0.05).

contribute to a common pool of AMR. The highest AMR levels in aquaculture were observed in economically vulnerable countries (i.e. LMICs), especially in Africa and South East Asia, which is consistent with the results of global AMR gene abundance found in sewage waters[40] and farmed terrestrial animals[41]. Higher AMR levels in LMICs can be linked to factors such as poorer sanitation systems or antibiotic misuse and highlight the need to establish regulations, controls and information systems in those countries[40,42,43].

In addition, we found that higher AMR levels of aquaculture-related bacteria were correlated with warmer temperatures, an association that has recently been observed amongst human clinical bacteria in the United States[35]. Although drivers behind this association are still unclear and are likely multi-factorial, these could include higher use of antimicrobials linked to increases in disease frequency at higher temperatures[44]. Current predictions suggest an increase in EIDs with global warming[15–19], which might pose further threats to food security, as aquatic animal diseases are one of the major limiting factors to the expansion of the aquaculture industry. Here we show that warmer temperatures almost always result in higher mortalities of infected aquatic animals, regardless of the type of animal cultured (shellfish, crustaceans or fish). As our results are based from experimental infections, further validation from field observations is required to reduce uncertainty. Whether or not higher temperatures select for increased pathogen virulence is still under discussion[21,45]; yet, extreme thermal increases are known to cause stress and compromise immune systems in most aquatic species, making them more vulnerable to infections[46,47]. Previous research found, for example, that an increase in severe disease outbreaks in aquatic species at lower latitudes could be partly due to warmer temperatures and higher nutrient contents[48]. Since antibiotic treatment decision is generally made at the onset of a disease outbreak, antibiotic use is unlikely to increase as a direct consequence of higher mortalities. However, increasing fish health challenges (i.e. increased mortalities and more frequent outbreaks) is likely to result in increases in the use of anti-microbial drugs[49], posing further health concerns as aquaculture-derived MAR is correlated to a country's climate vulnerability. This shows that countries struggling the most to respond and adapt to climate change will also face the highest risk of AMR.

Altogether, these findings emphasise the need for urgent coordinated national and international interventions to limit antimicrobial use and the global spread of AMR[50,51]. In some countries, the use of antimicrobial drugs in animal production exceeds human medicine use[6], significantly contributing to the emergence and spread of antibiotic resistant bacteria[41], one of the

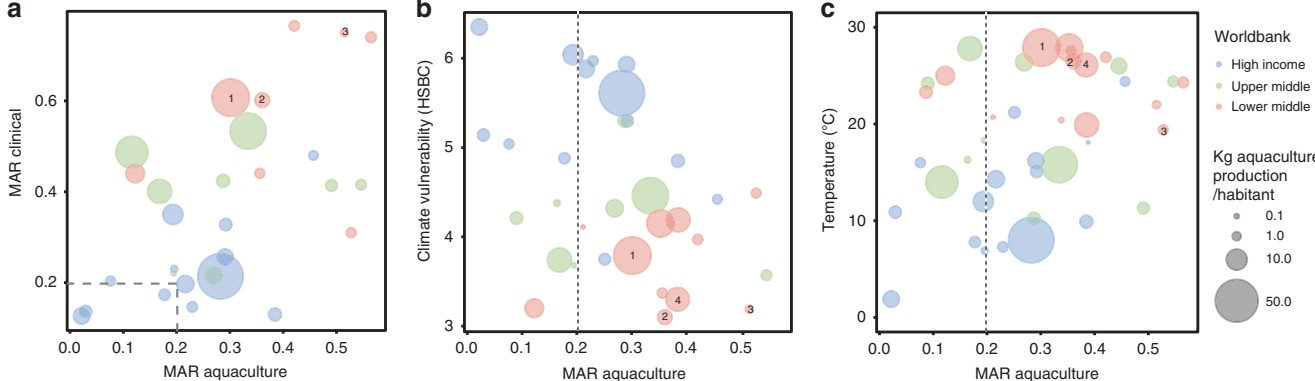

**Fig. 3 Correlations between MAR calculated from aquaculture-related bacteria and human clinical bacteria, temperature, and countries' climate vulnerability.** Pearson correlations (two-sided test) **a** MAR from human clinical bacteria ($n = 29$, $P$-value < 0.001), **b** HSBC climate vulnerability index ($n = 32$, $P$-value = 0.020) and **c** temperature ($n = 40$, $P$-value 0.10). Bubbles sizes are proportional to national aquaculture production standardised by the total country human population. The colours indicate different Worldbank categories (High income, Upper-middle income and Low-middle income). 1: Vietnam, 2: India, 3: Pakistan, 4: Bangladesh displayed simultaneously the highest levels of clinical and aquaculture MAR and are among the ones exposed to the highest climatic vulnerability and temperatures rises. See the Methods section for details.

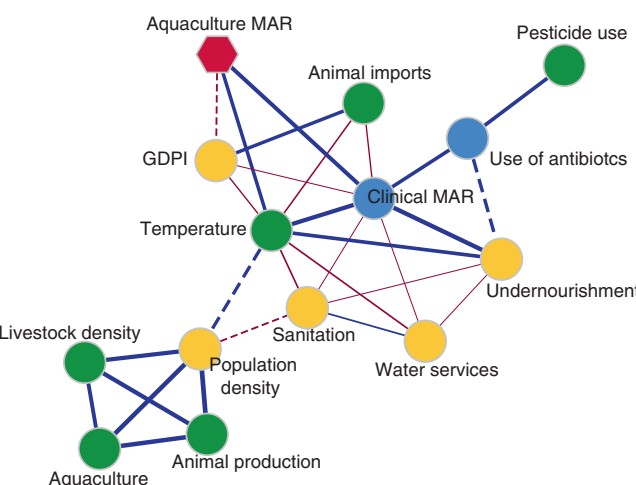

**Fig. 4 Pearson correlation network between all the simple studied variables.** Significant correlations ($P$-value < 0.05) are displayed with solid lines, whereas correlations ($r > 0.30$) nearing statistical significance ($0.10 > P$-value > 0.05) are shown in dashed lines. Edge weight is proportional to the correlation coefficient ($r$), with line width increasing with higher correlation values.

major threats of the twenty-first century according to the World Health Organization. Despite decreasing aquaculture antimicrobial use in recent years, partly due to the banning of growth promotion treatments in many countries, information of detailed antimicrobial use remains scarce, hindering assessment of their human, animal and environmental risks[52,53]. Out of 60 different antimicrobial drugs currently used in aquaculture, 40 are classified as critically important or highly important by the World Health Organization, highlighting the urgent need for antibiotic regulation reinforcement, control and reporting in aquaculture[53–56]. About 80% of antimicrobials administered through feed to aquatic farmed animals disseminate to nearby environments (water and sediment) where they remain active for months at concentrations allowing selective pressure on bacterial communities and favouring AMR development[22,57,58]. Aquatic environments, often contaminated with AMR from terrestrial effluents, are considered hotspots for AMR bacteria and AMR genes acting as sources of horizontal gene transfer to the human and animal resistome

(all AMR genes found in the human/animal microbiome)[29,59]. Therefore, better management of crops, animal production systems and sewage is required to avoid cross-contamination between terrestrial and aquatic environments. Many strategies have been proposed to limit antimicrobial use in aquaculture, ranging from better disease surveillance and management to improving animal fitness or ecosystem resilience[60,61]. In this context, some practices contributing directly to AMR emergence or ecosystem cross contamination should be avoided and more sustainable practices encouraged. For example, using antibiotics as growth promoters in animal feeds is still common practice in several countries and directly contributes to the emergence of AMR[62]. Integrated farms, which combine fish/shellfish rearing with farming livestock (e.g. pigs or chickens), despite being considered sustainable, may favour the emergence and spread of antimicrobial resistance[63]. In contrast, an ecosystem approach to aquaculture (EEA) and integrated agriculture-aquaculture such as prawn/fish-rice farming, may improve ecosystem resilience, increasing disease resistance in farmed animals and decreasing dependence on veterinary drugs, whilst providing social benefits such as local food security and higher incomes[64,65].

Several sustainable solutions exist to minimise antimicrobial use in aquaculture by increasing animal welfare and disease resistance. For instance, vaccination, a highly investigated alternative, has proven very effective in reducing antimicrobial consumption in Norwegian salmon farms. However, it is often too expensive and unsuitable for other types of aquaculture species such as those farmed in developing countries or where animals are infected by multiple opportunistic pathogens[66]. The use of food supplements that maximise fish growth and feeding efficiency whilst enhancing their immune system and thus disease resistance has gained considerable attention in disease prevention of aquatic farmed animals[67,68]. These techniques have the advantage of often being affordable for small-scale fish farmers, improving feeding efficiency, which is currently a major limiting factor for aquaculture[69] and reducing the use of drugs. Probiotics (live microorganisms) and bioactive plants are amongst the most studied feed supplements, with a growing amount of literature showing their beneficial effects on animal growth and immunity[67,68]. Such alternatives are already widely used in small-scale farms of SE Asia[70,71], showing their potential as affordable sustainable alternatives to fish health challenges. Our findings, together with the evidence that restricting AMU in food-producing animals decreases global AMR[72], highlight the

importance of shifting towards sustainable infectious disease prevention strategies in animal production systems.

## Methods

**Literature research strategy**. We systematically searched all peer-reviewed journal articles using Web of Science and Google scholar up to 1 March 2019 that investigated (1) mortalities from cultured aquatic animals due bacterial infections (dataset 1) and (2) AMR from aquaculture environments (dataset 2). Since AMR changes over time, we only retained articles on this subject published within the last 10 years. We performed two independent literature searches for each of the subjects following the PRISMA (preferred reporting items for systematic reviews and meta-analyses) guidelines and research synthesis norms[73,74] (Supplementary Figs. 2, 3). The following keyword combinations were used: (1) (aquaculture* OR farm* OR rear*) AND (fish OR shrimp OR shellfish) AND (mortality OR outbreak OR infection) AND (Aeromonas OR Edwardsiella OR Flavobacterium OR Streptococc* OR Vibrio OR Yersinia) and (2) (antimicrobial or antibiotic) AND (resistance OR susceptibil*) AND (aquaculture OR fish OR shrimp OR shellfish).

These searches produced a total of 3,526 records for dataset 1 and 4,512 records for dataset 2 that were filtered in a three-stage process (Supplementary Figs. 4, 5). After removal of duplicates, issued from combining several database searches, title and abstract of the remaining records (2,458 for dataset 1 and 2,556 for dataset 2) were scanned for relevance in the studied topics. Then, the full-texts of the retained articles (837 for dataset 1 and 697 for dataset 2) were assessed (Supplementary Figs. 4, 5).

**Inclusion criteria and data extraction**. Dataset 1: Only research articles where an experimental infection was performed with a clear identified protocol were considered. Natural outbreaks were not considered due to the difficulty of determining (1) whether a previous treatment (e.g. vaccine or antibiotic) was applied, (2) exact temperature during the duration of the outbreak and (3) whether the outbreak was uniquely caused by one clearly identified bacterial pathogen. All selected studies met the following criteria: (1) experimental infections were performed with pure bacterial cultures previously characterised to species level, (2) dose of infection and mode of infection were clearly identified, (3) the life stage of the organism infected was reported, (4) temperature during the duration of the outbreak was clearly reported and constant ($\pm 1$ °C), (5) the animal mortality was reported as % and (6) aquatic infected animals were not exposed to any substance or stress that might have interfered with the mortality outcome. When a study included several experiments under different temperatures, host species or pathogen species, we considered them distinct observations. Following all the aforementioned criteria, we obtained a dataset containing 582 observations extracted from 273 studies (Supplementary Figs. 4, 6, Supplementary data 1). For each of the observations we extracted the following data: pathogen and host taxonomy (species, family and phylum), host developmental stage (larvae, juvenile, adult), country, temperature of the infection, cumulative mortality, mode of infection (injection or immersion) and infective dose (CFU/fish if injected or CFU/mL if challenged by immersion).

Dataset 2: Only research articles reporting antimicrobial resistance of bacteria isolated directly from the aquaculture environment (cultured animals recovered at the farmed site, water or sediment) were considered. Articles reporting antimicrobial resistance of isolated bacteria from cultured animals recovered in any other site than the farming environment, such as retail markets or imported products, were not included to avoid a bias introduced by potential contamination during transport. All selected studies met the following criteria: (1) antimicrobial activity of bacterial isolates was reported for at least three antibiotics, (2) at least the bacterial genus was identified in order to be able to disregard susceptibilities to antibiotics for which they are naturally resistant (Supplementary Table 17) and (3) bacteria studied were known as pathogenic for aquatic cultured animals. Since *Pseudomonas* species are known to present numerous intrinsic AMR, they were excluded from the analysis in order to avoid biased results. For the calculation of the countries' MAR indices, we established a minimum requirement of 30 bacterial isolates. This led to a dataset that contained antimicrobial resistances of 11,274 isolates extracted from 187 studies (Supplementary Figs. 5, 7, Supplementary data 2). For each of these studies the following information was extracted: country of the study, bacterial species or genus, source of isolation (host species or type of farm), number of antibiotics tested and number of resistant isolates.

**Mortality vs temperature data analysis**. The dataset was divided into subsets according to host thermal range (tropical-subtropical and temperate), host phylum (arthropods, molluscs and chordates) and pathogen family. In total, we obtained 12 subsets that were host- (phylum) and pathogen- (family) specific and two general subsets that combined different hosts with same thermal range (tropical-subtropical and temperate) (Supplementary Fig. 6). Nested linear regression and linear mixed models were constructed to examine the relationship between mortality of infected aquatic animals and temperature. Fixed effects included temperature, life stage, mode of infection and infective dose (log-transformed). Study ID was included as random effect unless its inclusion resulted in model singularity. Random effects on host and pathogen taxonomy were included to account for variation in the response variable related to multiple observations on similar taxa.

Host family and pathogen species were included as random effects in the pathogen specific datasets, while nested host (phylum/family) and pathogen taxonomy (family/species) were also included in the general datasets. Akaike's Information Criterion for smaller sample sizes (AICc) was used to assess the explanatory value and parsimony of each model. The difference in AICc values between each model and the best fitting model with the lowest AICc ($\Delta$AICc) was used to determine the strength of each model. Akaike weights ($w_i$), which determine the weight of evidence of each model relative to the set of candidate models, were then used to select the model with the best fit (model with the highest weight)[75]. Firstly, we constructed and compared models with temperature as the only fixed effect to evaluate if the use of random effects was justified. Secondly, nested models were constructed on the previously selected fixed or mixed model to select for the fixed effects that resulted in the best model. All models were built using the lme4 package for R software (R version3.6.0.) and selection was performed using the function model.sel from the MumIn package for R.

**Antimicrobial resistance data analysis**. The MAR index was calculated for individual bacterial isolates (i.e. strains or species) when possible or for groups of isolates (same bacterial genus) as the ratio between the number of resistant bacterial isolates and the number of total combinations tested (number of antibiotics * number of isolates tested)[36]. A MAR index for each country was then obtained as the mean of all MAR indices obtained for that country and weighted by the number of isolates used to compute them.

Pearson correlation analyses between the country MAR index (from aquaculture data) and several environmental, socioeconomic and health indicators were performed to investigate sources that might affect AMR using the cor.test function from the R package stats. Indicators were collected for the 40 countries for which we computed the MAR index for 2016, when available. GDP per capita, HDI, population, prevalence of undernourishment, % of people using at least basic sanitation services and % of population using at least basic water services were downloaded from Worldbank Open Database (https://data.worldbank.org). Pesticide use, animal production, livestock density and animal trade data were downloaded from the FAOSTAT webpage (http://www.fao.org/faostat). Aquaculture production was obtained from the WAPI Aquaculture Production module developed by the FAO (http://www.fao.org/fishery/statistics/software/wapi/en). Animal production, animal trade and aquaculture production were normalised by the area ($km^2$) of the country. Environment performance index (EPI), an index that evaluates environmental health and ecosystem vitality, was collected from https://epi.envirocenter.yale.edu.

The human clinical use of antibiotics and the AMR (number of isolates tested and % of resistance) of *Escherichia coli* to aminoglycosides, third generation cephalosporins and fluoroquinolones were obtained from https://resistancemap.cddep.org/. *E. coli* AMR to the three classes of antibiotics was used to calculate an index of clinical MAR. Also, the index of climate vulnerability used in the present study was that defined and computed by HSBC[38]. This index is calculated assigning equal weights (25%) to four indicators: (1) physical impacts (average temperature and changes, water availability and probability of extreme weather events), (2) sensitivity to extreme events (number of fatalities, damage costs and number of people affected), (3) energy transition risks (diversification of exports, energy and GDP away from fossil fuels) and (4) a country's potential to respond to climate change, which includes data on the country financial resources (GDP per capita, debt, equity risk and sovereign wealth) and national governance (tertiary education, rule of law, corruption and inequality). In order to obtain country temperatures representative of the sites from which the MAR indices were calculated, regional average of annual temperatures (1991–2016, https://climateknowledgeportal.worldbank.org) for each study were collected using the geographical coordinates of the studied sites. For studies in which sites were not reported, the mean annual country temperature was collected. Temperatures for each country were then calculated computing the weighted mean of the regional study temperatures by the total number of isolates analysed in the study.

In order to investigate the presence of underlying co-founding factors in the variability of aquaculture-derived MAR indices, we investigated the Pearson correlations between each pair of simple variables. Then a correlation network was built with Cytoscape v.3.7.2 when correlation between two variables was statistically significant ($P$-value < 0.05). Variables with correlations $r \geq 0.30$ and $P$-values nearing significant cut-off ($P$-value < 0.07) were also included in the network and graphically identified. Composite variables (i.e. computed through the combination of several variables such as the HDI or CVI) were excluded to simplify data interpretation. We then performed a multiple regression model, with the variables correlated to MAR aquaculture indices in order to identify their contribution as well as their interaction to the aquaculture-derived MAR index. Nested models with the variables and their interactions were built and the model explaining the highest variance ($R^2$) was selected. However, since the explanatory variables were highly correlated between them ($r > 0.6$) and multicollinearity reduces the precision of the coefficient estimates of the model, we used Lindeman, Merenda and Gold (LMG) relative importance method in linear regression (package relaimpo for R), to calculate the percentage of relative importance of each of the model variables with their 95% bootstrap confidence intervals (1000 permutations).

**Reporting summary**. Further information on research design is available in the Nature Research Reporting Summary linked to this article.

## Data availability

All data generated from this study (meta-data used to study the mortalities of infected aquatic animals in relationship to temperature and meta-data used to calculate the aquaculture-derived MAR) are available on the public repository DRYAD: https://doi.org/10.5061/dryad.dv41ns1tr

## Code availability

Source codes of the models are available on the public repository DRYAD: https://doi.org/10.5061/dryad.dv41ns1tr

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

## Acknowledgements

This work was funded by the French National Research Institute for Development (IRD), by the French Ministry of Agriculture and Food (Ecoantibio-2018-196) as well as by the LabEx CeMEB with an ANR "Investissements d'avenir" program (ANR-10-LABX-04-01). Authors would like to thank J. L. Gozlan for language revision.

## Author contributions

M.R., R.E.G., S.S., D.C., J.-C.A., H.V., L.P., E.P., M.C., and S.V.H. conceived the study. M.R. collected and analysed the data. M.R. and R.E.G. wrote the article. All authors reviewed and commented to the final version.

## Competing interests

The authors declare no competing interests.
