## [Peer Review File · Nature Communications]

Reviewers' comments:

Reviewer #1 (Remarks to the Author):

Dear Editor and authors,

I have read the paper titled "Aquaculture at the crossroads of global warming and antimicrobial resistance: shifting towards sustainable solutions". I think it is well written and the subject matter is critically important given the predicted expected growth in the aquaculture worldwide. I think there are four issues that need to be addressed before this manuscript can be considered for publication. First, the implied goal of the study appears to be to link the use of antibiotics to climate change. The authors set the stage to do this but need to make a stronger link between the total mortality associated with bacterial diseases, which is predicted to increase with increase temperatures, and the use of antibiotics. If they can make a link between an increase in bacterial disease frequency and temperature, that would make a stronger case for the potential for increased use of drugs (I am sure this is the case but I am not sure it is documented very well in the literature). Instead of taking the reader down this path the authors make the claim that the risk of AMR is due to exposure of bacteria to antibiotics at high temperatures, but no data are given to support this idea, and the paper cited does not provide information on the mechanism for the trend observed between AMR and temperature. Secondly, the authors often reference papers incorrectly (either they over interpret the conclusions of the paper or they cite a review paper that does not provide empirical data for the statements made). Third, the authors include information on viral diseases in this manuscript, which distracts the reader from the issue of antibiotic treatments. Lastly, the authors need to remove the plant study as the they provide no data for the efficacy of plant based medicines in aquaculture. I could not tell if the studies they were referring to had proper controls or how they monitored improved health. The latter is a study on its own and the methods and analysis of their data need to be evaluated before this can be published. Currently there is insufficient information given in the manuscript and the supplemental document to assess the authors conclusions on the plant therapy.

I have included my comments in the word version of the manuscript for your review.

Reviewer #2 (Remarks to the Author):

The authors have done a laudable amount of work to generate the interesting models and conclusions presented here. The paper is in general very well written, but in some instances English needs a bit of work with the occasional odd turn of phrase or handling of plural or singular terms.

The growing need to produce protein in aquatic systems at a time of increasingly problematic climate change is a very important challenge. The central conclusion from this study is that a meta-

analysis of the literature indicates that increased water temperature results in increased health challenges. There is indeed quite a body of literature now indicating that increasing water temperature and acidification [which does not appear to have been considered here] due to climate change is associated with more frequent and problematic fish diseases caused by viruses, bacteria or parasites, as well as increased frequency of harmful algal blooms that can poison production areas, not considered in this review. The authors review indicates that bacteria from warmer lower income areas of production have more frequent resistance than higher income countries, in agreement with what is observed with clinical bacteria. A conclusion from these observations could be that antimicrobial use will have to increase in response to fish health challenges, and that this will result in more antimicrobial resistance development in these food production systems. The authors propose that introducing medicinal plants into these systems would be effective at keeping fish healthy, thus reducing the need for antimicrobial use. They also suggest that globally a range of strategies would need to be deployed to confront this challenge, perhaps medicinal plants, but also vaccines, breeding for resistance, better location of production areas and management of animal density, improved husbandry in general as and where appropriate. There has to be better management of terrestrial crop and animal production systems so that nutrients don't cross the terrestrial-aquatic interface causing eutrophication and a suite of consequent problems. Overall, the observations are as would have been expected, but the global breadth presented in this meta-analysis is compelling and should be of broad interest.

Reviewer #3 (Remarks to the Author):

This is a substantive piece of work drawing important and novel conclusions that the policy and research communities need to react to. Inevitably, there are weaknesses and I invite you to consider the following comments.

General

Your arguments are compelling and rely on drawing multiple evidence strands together. You have not unreasonably adopted rapid review methodologies to make this task tractable (e.g. no protocols, high specificity searches, no critical appraisal, no assessments of publication bias). I do not have a problem with this approach BUT it would be worth acknowledging where this increases uncertainty in the discussion at least. Reference to evidence synthesis norms would be helpful in encouraging future high quality synthesis and nuanced judgement when adopting rapid review methods.

Inclusion of data sets and code to replicate analysis should be a pre-requisite for a publication of this type in Nat Comms. Please annotate the code to make as useable as possible for others who may wish to extend the analyses or adopt your methods in other contexts

I struggled to understand the significance of the MAR index in the introduction (although it was clear when I read the methods). Consider revising for the less microbially astute amongst the readership. This applies to the discussion of AMR in the introduction too. Maybe trial comprehension on some PGR from another department prior to finalisation?

line 91. remove the p value in brackets- we can see this in the table and the important information about the size of the effect is conveyed in the next sentence.

131 to 158.. This is very clear and important science. I am impressed. However, it is worth pointing out that the causal inference is uncertain here and potentially, considering how this could be addressed by future research focused on the high risk LMICs?

Replace table two with a graph with error bars or confidence intervals?

line 225 typo widespread should read spread?

Gavin B Stewart

University of Newcastle

Reviewer #1

General comment 1: *First, the implied goal of the study appears to be to link the use of antibiotics to climate change. The authors set the stage to do this but need to make a stronger link between the total mortality associated with bacterial diseases, which is predicted to increase with increase temperatures, and the use of antibiotics. If they can make a link between an increase in bacterial disease frequency and temperature, that would make a stronger case for the potential for increased use of drugs (I am sure this is the case but I am not sure it is documented very well in the literature). Instead of taking the reader down this path the authors make the claim that the risk of AMR is due to exposure of bacteria to antibiotics at high temperatures, but no data are given to support this idea, and the paper cited does not provide information on the mechanism for the trend observed between AMR and temperature.*

General response 1: We would like to clarify that the goal of the paper was not to link the use of antibiotics to climate change but instead, as stated in our title, to show that “aquaculture is at the crossroads of global warming (i.e. temperature increase) and antimicrobial resistance”. We tried to show this by 1) studying some of the aquaculture health challenges that might arise with warmer temperatures (e.g. higher mortalities) 2) studying the state of antimicrobial resistance in aquaculture and 3) correlating antimicrobial resistance in aquaculture to socioeconomic variables. Our study clearly shows that among other factors, there is a significant correlation between multi-antibiotic resistance calculated indices from aquaculture-related bacteria and temperature (Table 2). In addition, we tested the association between low- and middle-income countries and MAR. As such, we have found that countries such as Vietnam,

India, Pakistan and Bangladesh simultaneously displayed the highest levels of clinical (human) and aquaculture MAR and were exposed to the highest climatic vulnerability and temperature, suggesting LMICs were regions most at risk for the combined action of global warming and AMR occurrence.

To study the effect of global warming on aquaculture, we focused on studying how mortalities associated with bacterial diseases are related to temperature. We agree with referee #1 that antibiotic use will not directly increase as a cause of higher mortalities, because the decision to treat with antibiotics is generally made prior to observing high mortalities. We however argue, as referee #2 stated, that antimicrobial use will have to increase in response to fish health challenges (which might not only include higher mortalities but also an increase in disease frequency), we have assessed this point in lines 220-223. We also agree that showing a clear increase in the frequency of aquaculture outbreaks in response to increasing temperatures is of the utmost importance. However, as mentioned by referee #1, there is a clear lack of published data to directly link a global increase in bacterial disease frequency to temperature. When we conceived the article, we tried to explore this subject consistently but failed to establish the association referee #1 proposed because of a lack of data in the literature.

Therefore, we acknowledge that this study shows an association between temperature and antibiotic resistance, but not a causality. At this stage, we can only speculate on the mechanisms involved, supporting referee #1 comment: “the underpinning mechanisms are likely to be multifactorial and could actually be related to the increase use of antimicrobials at warmer temperatures due to more bacterial infections in the population”. To address this concern, we have now modified the text throughout (lines 204-226) to unambiguously describe an association rather than a causal link, which was not the scope of our study. We have also strengthened the link between bacterial disease frequencies and temperature throughout the text, with the inclusion of additional references (FAO 2018, Bondad-Reantaso & Melba 2016), which report increases in bacterial disease in aquaculture farms as a result of increased temperature. These are expert reports and occasional field evidence, but there is a lack of systematic reporting of bacterial mortalities in aquacultures worldwide. Finally, since our paper submission, a study has been published in *Science* (Van Boeckel *et al.* 2019) showing similar trends between LMIC countries and MAR in farmed terrestrial animals (chickens, pigs). This reference has now been added in the discussion to strengthen our findings.

We hope that this general response will satisfy the general comments of referee #1, but we have provided specific text modifications aimed at addressing all specific comments in the point by point responses below.

References:

- Bondad-Reantaso & Melba. Acute hepatopancreatic necrosis disease (AHPND) of penaeid shrimps: Global perspective. SEAFDEC <http://hdl.handle.net/10862/3084> (2016).
- FAO. *Impacts on climate change on fisheries and aquaculture – Synthesis on current knowledge, adaptation and mitigation options*. Rome. (2018).
- Van Boeckel, T.P., et al. Global trends in antimicrobial resistance in animals in low- and middle- income countries. *Science* **365**, 1266 (2019).

General comment 2: *Secondly, the authors often reference papers incorrectly (either they over interpret the conclusions of the paper or they cite a review paper that does not provide empirical data for the statements made).*

General response 2: Following this comment, we reviewed in detail all cited publications and either modified our interpretation or replaced review references by source references as requested by reviewer. This can be found in our point by point responses.

General comment 3: *Third, the authors include information on viral diseases in this manuscript, which distracts the reader from the issue of antibiotic treatments.*

General response 3: We agree and have now removed all the information related to viral diseases.

General comment 4: *Lastly, the authors need to remove the plant study as the they provide no data for the efficacy of plant based medicines in aquaculture.* I could not tell if the studies they were referring to had proper controls or how they monitored improved health. The latter is a study on its own and the methods and analysis of their data need to be evaluated before this can be published. Currently there is insufficient information given in the manuscript and the supplemental document to assess the authors conclusions on the plant therapy.

General response 3: We have removed Box 1 on plant studies, as also requested by the Editor. However, considering that plant therapy is widely used in small-scale aquacultures and that it may constitute a sustainable alternative to antibiotics, we have added one sentence on this aspect in the discussion, based on published data (lines 274-276).

Point by point detailed responses to Referee #1' s comments:

- Comment Line 13: This suggests that aquaculture practices lead to all the MAR found. It is likely that AMR in aquaculture is acquired from multiple sources as everything ends up in the watershed.

Response: Agreed, we have now deleted “– derived” to widen the scope and include multiple sources.

- Comment Line 18: Could delete this part of the sentence as it is speculative and not supported by data.

Response: This part of the sentence has now been deleted.

- Comment Line 29: Add ‘husbandry’

Response: It now reads ‘good husbandry practice’

- Comment Line 34: Define a solution to what?

Response: It has now been clarified ‘a solution to the need for protein’

- Comment Line 43: Perhaps leave this pathogen out and include another bacterial agent as people should not use antibiotics to treat viral infections. Viral and parasitic infections often lead to secondary bacterial infections, which can then result in the use of antibiotics but this is a bit complicated to explain in the intro and I don’t think it is necessary.

Response: We agree and we have now replaced it with acute hepatopancreatic necrosis disease (*Vibrio parahaemolyticus*) and added a supporting reference.

- Comment Line 46: add 'And compete economically'

Response: We agree and have now added "to meet economic requirements and the demand for animal products".

- Comment Line 47: There are a lot of sources of antibiotics in the aquatic environment that are not from aquaculture so it is important not to over simplify the situation. If aquaculture stopped using antibiotics tomorrow, there would still be a significant amount of exposure to antibiotics if fish farms are around large cities and using surface water.

Response: We agree and made that point clearer. We added: "...aquaculture and land-derived contamination into watercourses ...".

- Comment Line 50: Maybe give the amount of deaths in SE Asia. It is higher.

Response: We have not been able to find this figure for the whole SE Asia, although we found it for Thailand (38,000/year). We nonetheless modified the text to reflect this comment. "Antimicrobial resistant (AMR) bacteria cause over 35,000 human deaths annually in the the USA, 33,000 in the European Economic Area, 58,000 in India and probably more in SE Asia and numbers are expected to rise²⁴⁻²⁵ due to rapid socioeconomic development and human population growth."

- Comment Line 56: Maybe explain in one sentence the other contributors to AMR risk (human waste water, animal and plant agricultural run-off, hospital effluent, etc...)

Response: We have now clarified this point accordingly. "Since aquatic environments are effective reservoirs of AMR bacteria²⁷ from other sources (e.g. human waste water, hospital effluents and animal and plant agricultural run-off), the direct..."

- Comment Line 59: This is minor when fish are on heat treated pellets. As farm become bigger there is a move towards more pelleted feeds. The paper you cite has many issues that make the conclusions questionable. They do not report resistant bacteria they report antibiotics in the fishmeal and ARGs but since fishmeal comes from wild fish sources it seems unlikely that raw fishmeal products would have so many antibiotics. The levels the authors report are also at the limit of most HPLC techniques. Regardless the paper is about antibiotics and ARCs more than resistant bacteria. I think the author's statement is misleading and it really is not necessary. The therapeutic use of antibiotics is really what they are trying to get at (address) with this paper. The focus should be there is use of drugs and with climate change it will increase (i.e. more disease more use of drugs).

Response: We agree with this comment and we have modified the sentence in order to reflect this point: "The aquaculture sector contributes to the AMR reservoir mainly by administering therapeutic and prophylactic antimicrobial treatments to animals but also to a minor extent with the use of non-antibiotic chemicals (e.g. disinfectants), which has been shown to increase AMR^{30,31}".

- Comment Line 63: The link to human health is logical but difficult to « prove » unless the authors have good references with empirical data they should include the word « may »

Response: We have added 'may' to reflect that level of uncertainty.

- Comment Line 65: This paper was in rodents and may not apply to fish. It is a bit misleading. There are plenty of valid reasons for not wanting AMR in aquaculture there is no need to include reasons that have not been demonstrated yet.

Response: We agree that the use of rodents as a model could have been misleading. However, we believe that it is an interesting novel aspect, which could benefit from being tested in fish. Therefore, we have now clarified the sentence accordingly “...but could also impact production itself by lowering drug efficacy^{23,32}, decreasing the animal’s immune system as seen in rats³³ and selecting more virulent strains...”.

- Comment Line 68: The paper cited is an ecological study so it did not look at the spread of AMR. It found an association between temperature and AMR. The mechanism for increase AMR in warmer temperature is likely multifactorial and could actually be related to the increase use of antimicrobials at warmer temperatures due to more bacterial infections in the population. I think the authors should modify this sentence and indicate that warmer temperatures have been associated with higher AMR rates in terrestrial bacteria and leave it at that.

Response: We agree that the association highlighted in the cited paper does not reflect a spread of AMR and as such we have modified the sentence to integrate this point: “...warmer temperatures have been associated with higher AMR rates in terrestrial bacteria, establishing a sombre...”

- Comment Line 81: At risk of what? please define

Response: This has now been clarified: “...environmental and socio-economic indicators to map countries or regions that are most at risk of seeing increases in AMR...”

- Comment Line 82: This should be published elsewhere on its own.

Response: We have now removed it from this paper.

- Comment Line 91: Were these laboratory studies or field studies or clinical trials?

Response: In this part, we only included studies that reported mortalities from experimentally infected animals (clinical trials). We initially collected information on both natural disease outbreaks and clinical trials. There were only a few natural outbreaks where temperature was reported, and it was often a temperature range. Furthermore, for natural outbreaks, it was not possible to control whether a previous treatment (vaccination or antibiotic administration) was applied that could have influenced the mortality outcome. Therefore, to avoid this potential bias, we decided not to include the natural outbreaks in our database. We have modified this sentence to clarify that studies are animal clinical trials.

“After screening the literature, we extracted data from a total of 273 studies to test the influence of temperature on the mortality of cultured aquatic animals experimentally infected with major bacterial pathogens...”

It has also been specified in the material and methods (lines 334-338) of the reviewed manuscript.

“Dataset 1: Only research articles where an experimental infection was performed with a clear identified protocol were considered. Natural outbreaks were not considered due to the difficulty of determining 1) whether a previous treatment (e.g. vaccine or antibiotic) was applied 2) exact temperature during the duration of the outbreak and 3) whether the outbreak was uniquely caused by one clearly identified bacterial pathogen.”

- Comment Line 94: If you are going to include viruses in your study you need to link the use of antibiotics with viral infections. The only time anyone should be using antibiotic with a viral infection is if there is a co-infection with bacteria. This does happen but I am not sure how often

it is published. Including the viruses here without anchoring them to the use of antibiotics makes it seem like the authors do not know that antibiotics are used for bacterial pathogens.

Response: We agree and we have decided to remove viruses from the manuscript for the purpose of clarity.

- Comment Line 96: This type of study design is not sufficient to demonstrate causality

Response: We agree, 'associated' is a better word.

- Comment Line 95: The models did not take into account whether there were any interventions given (specifically vaccines prior to infection and antibiotic treatments after mortality started). These could confound your results. Further it is not clear how the authors accounted for the clustering by study (I assume they included this as a random effect? but I could not find mention of it).

Response: Since all observations came from experimental infections, there were no previous interventions (vaccines or antibiotics) that could affect the mortality outcome. We have modified the text to clearly show when study id was included as random effect. This can be seen in line 95 of results, table 1, supplementary tables 1-15 and the materials and methods section (402-403).

- Comment Line 115: This is nice information. I am not sure how the level of mortality is linked to the use of antibiotics. Typically, once fish start dying they are treated so the level of mortality may not affect the decision to treat once you get past a certain threshold.

Response: We have removed the part regarding viruses and in the revised version we only include the bacterial example. Regarding the second point highlighted, we agree that decision on antibiotic treatment is made before high mortalities are observed, and therefore there is not a direct causality between increase of mortality and use of antibiotics. However, we agree with the point made by referee #2 that "antimicrobial use will have to increase in response to fish health challenges", and that there might be an association between higher temperatures, more severe diseases (i.e. higher mortalities), possibly an increase of disease frequency (although this remains to be studied in depth) and therefore an increase in the use of antibiotics to respond to the aforementioned challenges. This point, has been addressed and discussed in the discussion section in lines 204-226.

- Comment Line 143: This is an ecological study that does not provide conclusive evidence of the mechanism by which temperature is associated with AMR. In fact, it could just be that you have more infections at higher temperatures and therefore you treat more.

Response: We agree with this comment. With our current understanding, both mechanisms could take place. We have removed the sentence and citations from the results section and addressed this accordingly, including the point raised, in the discussion section in lines 206-208. "Although drivers behind this association are still unclear and are likely multi-factorial, these could include higher use of antimicrobials linked to increases in disease frequency at higher temperatures⁴⁴".

- Comment Line 149: As a veterinarian in Asia I see a lot of AMR in fish bacterial isolates but the patterns are more suggestive of resistance from human use of antibiotics (resistance to drugs that are too expensive for small scale farmers to use) so your finding makes sense. I think there is over use of some drugs in aquaculture (mostly to the cheaper antibiotics) and this is also likely driving resistance to specific families of products.

Response: Some of the co-authors have been working in Asia for the last 10 years and have drawn similar conclusions.

- Comment Line 152: What is the difference between total use of clinical antibiotics and the statement above.

Response: The above statement refers to the MAR from clinical bacteria. However, we acknowledge that the sentence was not clear and we have modified it to reduce any ambiguity. A strong positive correlation was found between human clinical MAR (MAR calculated from patient isolates³⁷) and aquaculture-derived MAR indices (Table 2, Fig. 3a), but no correlation was found between aquaculture MAR indices and the use of clinical antibiotics (antibiotics sold in retail and hospital pharmacies for human consumption³⁸) (Table 2).

- Comment Line 166: It is possible that there was confounding between predictors and these univariate analyses could not account for this and any chance of doing a more model with multiple variables in it? It is also possible that your estimate of MAR is biased because resistance to antibiotics is not published evenly across all countries.

Response: It is likely that the co-variables (e.g. animal production, animal trade, aquaculture production, reported pesticide use, % of undernourishment etc.) provided at country level are less variable than the estimates of MAR, which can indeed be limited by the number of reports, especially in the countries where there are few studies. This point is raised in the result section (lines 125-128). “It is however important to highlight that, despite our best efforts in gathering a global database, the calculated aquaculture-derived MAR indices might be limited by the uneven report of antimicrobial resistance between countries”.

Since we only want to show a possible association and not a causal link between several parameters, we have decided not to use more complex models, and have retained correlation coefficients. We have, however, included 95% interval confidences in the reviewed manuscript.

Comment Figure 3 caption: Human or fish clinical bacteria?

Response: This has now been clarified: “...from human clinical bacteria...”

Comment Line 196: *E. Coli* from humans?

Response: This has now been clarified: “...between MAR indices from aquaculture and human clinical bacteria...”

Comment Line 216: Higher temperatures usually results in higher replication rates of pathogens. If the temperature is within the normal range of the fish species it will not cause immune suppression. Note not all species have the same temperature range so this statement is over simplified.

Response: We agree with this comment, but our study is placed in the context of climate change where temperatures are not expected to be within their ‘normal range’ and where peaks of extreme heat are expected to increase in frequency. We have thus clarified the sentence accordingly: “...yet, extreme thermal increases are known to cause stress and compromise immune systems in most aquatic species...”.

Comment Line 225: These are review articles and you should assess and cite the primary sources.

Response: Since the second part of this sentence is based on our results (correlation between MAR from aquaculture-related bacteria and climate vulnerability), we deleted the references to the review articles highlighted, and we have not cited primary articles here.

Comment Line 230: This source (6) is about predicting the amount of drugs used in animal production not about the emergence and spread of resistant bacteria.

Response: We have moved this reference to the first part of the sentence and added the new reference “Van Boeckel, T.P., Pires, J., Silvester, R., Zhao, C., Song, J., Criscuolo, N.G. *et al.* Global trends in antimicrobial resistance in animals in low- and middle- income countries. *Science* **365**, 1266 (2019).

Comment Line 241: This is a review article. Original papers should be read, assessed for scientific rigor and cited. Many papers on this topic do not provide empirical data for their conclusions.

Response: We have read the source references and other research articles on the subject and have cited them accordingly.

Samuelsen, O.B., Torsvik, V., Evik, A. Long-range change in oxytetracycline concentration and bacterial resistance towards oxytetracycline in a fish farm sediment after medication. *Sci. Tot. Env.* **114**, 25-36(1992).

Tamminem, M., et al. Tetracycline resistance genes persist at aquaculture farms in the absence of selection pressure. *Environ. Sci. Technol.* **45**, 386-391(2011).

Comment Line 249: This is an awkward sentence

Response: The sentence has now been modified: “In this context, some practices contributing directly to AMR emergence or ecosystem cross contamination should be avoided and more sustainable practices encouraged.”

Comment Line 254: This paper just talks about antibiotic use patterns not about how this contributes to the emergence of AMR.

Response: We have now replaced the reference with: Meek, R.W., Vyas, H., Piddock, L.J.V. Nonmedical uses of antibiotics: time to restrict their use? *Plos Biol.* **13**, e1002266 (2015).

Comment Line 257: Farmers could still apply antibiotics to these systems

Response: We agree, this is why we said “decreasing dependence on veterinary drugs” and not “avoiding”. However, if such systems increase disease resistance and improve ecosystem resilience, it is likely that the need for antibiotics may also decrease to some extent. We have included a “may”, in order to be more cautious with the statement.

“In contrast, an ecosystem approach to aquaculture (EEA) and integrated agriculture-aquaculture such as prawn/fish-rice farming, **may** improve ecosystem resilience, increasing disease resistance in farmed animals and decreasing dependence on veterinary drugs, whilst providing social benefits such as local food security and higher incomes^{64,65}”.

Comment Line 270: This is not necessarily true. Feed supplements do not reduce feed input you still need to give the fish the building blocks to grow.

Response: We changed the sentence to be more accurate. “These techniques have the advantage of often being affordable for small-scale fish farmers, improve feeding efficiency...”

Comment Line 272: Hazardous drugs in this context makes no sense. If you treat with a medication it is not hazardous.

Response: We have removed the word 'hazardous'.

Comment Box 1: This should be published separately. Not sure where the data are?

Response: This box has now been removed, and a sentence to illustrate the widespread use of plants and their disease prevention potential has been added in the discussion section (line 270-273).

Comment Line 441: Human or aquaculture?

Response: Human. This has now been clarified.

Reviewer #2:

The authors have done a laudable amount of work to generate the interesting models and conclusions presented here. The paper is in general very well written, but in some instances English needs a bit of work with the occasional odd turn of phrase or handling of plural or singular terms.

The growing need to produce protein in aquatic systems at a time of increasingly problematic climate change is a very important challenge. The central conclusion from this study is that a meta-analysis of the literature indicates that increased water temperature results in increased health challenges. There is indeed quite a body of literature now indicating that increasing water temperature and acidification [which does not appear to have been considered here] due to climate change is associated with more frequent and problematic fish diseases caused by viruses, bacteria or parasites, as well as increased frequency of harmful algal blooms that can poison production areas, not considered in this review. The authors review indicates that bacteria from warmer lower income areas of production have more frequent resistance than higher income countries, in agreement with what is observed with clinical bacteria. A conclusion from these observations could be that antimicrobial use will have to increase in response to fish health challenges, and that this will result in more antimicrobial resistance development in these food production systems. The authors propose that introducing medicinal plants into these systems would be effective at keeping fish healthy, thus reducing the need for antimicrobial use. They also suggest that globally a range of strategies would need to be deployed to confront this challenge, perhaps medicinal plants, but also vaccines, breeding for resistance, better location of production areas and management of animal density, improved husbandry in general as and where appropriate. There has to be better management of terrestrial crop and animal production systems so that nutrients don't cross the terrestrial-aquatic interface causing eutrophication and a suite of consequent problems. Overall, the observations are as would have been expected, but the global breadth presented in this meta-analysis is compelling and should be of broad interest.

General response: Many thanks for your encouraging comments. Some of the points raised have now been included as part of the discussion in lines 220-223, 244-246.

We have also re-checked the English throughout all the manuscript, with special emphasis on plural and singular terms.

Reviewer #3:

This is a substantive piece of work drawing important and novel conclusions that the policy and research communities need to react to. Inevitably, there are weaknesses and I invite you to consider the following comments.

General comment 1: Your arguments are compelling and rely on drawing multiple evidence strands together. You have not unreasonably adopted rapid review methodologies to make this task tractable (e.g. no protocols, high specificity searches, no critical appraisal, no assessments of publication bias). I do not have a problem with this approach BUT it would be worth acknowledging where this increases uncertainty in the discussion at least. Reference to evidence synthesis norms would be helpful in encouraging future high quality synthesis and nuanced judgement when adopting rapid review methods.

Response: We have modified and increased the length of our 'material and methods' section in order to better clarify how the literature research was performed and the critical appraisal used for each of the datasets (lines 336-371).

We have also acknowledged in several parts of the manuscript (e.g. results lines 125-128 and discussion lines 213-215) the limits of our data and have included a reference that discusses the limits, challenges and perspectives of research synthesis and meta-analysis (line 320).

Gurevitch, J., Koricheva, J., Nakagawa, S., Stewart, G. Meta-analysis and the science of research synthesis. *Nature* **555**, 175-182.

General comment 2: Inclusion of data sets and code to replicate analysis should be a pre-requisite for a publication of this type in Nat Comms. Please annotate the code to make as useable as possible for others who may wish to extend the analyses or adopt your methods in other contexts.

Response: Both the annotated R code used to build the models and the corresponding raw data (Source_file_Part1: relationship between mortality and temperature and Source_file_Part2: data used to calculate the aquaculture-derived MAR at country level) have been submitted along with the revised version. Data and code have also been submitted to the public repository Dryad and are currently and can be accessed through the following link: https://datadryad.org/stash/share/JK183pVtScGkNSiRyj4B9LMpRDQzVifGDgEVF5pUR_o

General comment 3: I struggled to understand the significance of the MAR index in the introduction (although it was clear when I read the methods). Consider revising for the less microbially astute amongst the readership. This applies to the discussion of AMR in the introduction too. Maybe trial comprehension on some PGR from another department prior to finalisation?

Response: We have provided further information regarding antibiotic resistant terminology (AMR and MAR), especially in the introduction, in order to facilitate comprehension for people not familiar with the topic. We have also introduced a brief definition of MAR index in the introduction and slightly modified it in the M&M section.

- Comment line 97: remove the p value in brackets- we can see this in the table and the important information about the size of the effect is conveyed in the next sentence.

Response: This has been removed as suggested.

- Comment lines 142 to 173: This is very clear and important science. I am impressed. However, it is worth pointing out that the causal inference is uncertain here and potentially, considering how this could be addressed by future research focused on the high risk LMICs?

Response: We agree with the comment. In this part of the article, we chose to present correlation analyses, which show associations, rather than models drawing causal inferences because of the potential bias in our variable (MAR from aquaculture-related bacteria) and of the lack of understanding of the mechanisms that could drive the observed associations. We discuss the potential limits of our data (e.g. different sampling sizes between countries) in the results (125-128).

Replace table two with a graph with error bars or confidence intervals?

Response: 95% confidence intervals of each correlation have been added to table 2. However, we have kept the table since we consider that a table better summarizes the results. Moreover, some of the most important correlations are displayed in figure 3.

Comment line 228: typo widespread should read spread?

Response: This has now been corrected.

Reviewers' comments:

Reviewer #1 (Remarks to the Author):

Dear Editor

I think the authors improved the manuscript significantly, but they need to control for confounding in the analysis they present in table 2 before they can make strong statements about MAR (in the discussion. Besides general comment 1 the authors have addressed all of my concerns relatively well. I enclose a few comments in text boxes in the article itself and a file with my comments to general comment 1.

Dear Editor

I have made my comments in bold below. I think the authors improved the manuscript significantly, but they need to control for confounding in the analysis they present in table 2 before they can make strong statements about MAR. Besides general comment 1 the authors have addressed all of my concerns relatively well.

Reviewer #1

General comment 1: First, the implied goal of the study appears to be to link the use of antibiotics to climate change. The authors set the stage to do this but need to make a stronger link between the total mortality associated with bacterial diseases, which is predicted to increase with increase temperatures, and the use of antibiotics. If they can make a link between an increase in bacterial disease frequency and temperature, that would make a stronger case for the potential for increased use of drugs (I am sure this is the case but I am not sure it is documented very well in the literature). Instead of taking the reader down this path the authors make the claim that the risk of AMR is due to exposure of bacteria to antibiotics at high temperatures, but no data are given to support this idea, and the paper cited does not provide information on the mechanism for the trend observed between AMR and temperature.

General response 1: We would like to clarify that the goal of the paper was not to link the use of antibiotics to climate change but instead, as stated in our title, to show that “aquaculture is at the crossroads of global warming (i.e. temperature increase) and antimicrobial resistance”. We tried to show this by 1) studying some of the aquaculture health challenges that might arise with warmer temperatures (e.g. higher mortalities) 2) studying the state of antimicrobial resistance in aquaculture and 3) correlating antimicrobial resistance in aquaculture to socioeconomic variables. Our study clearly shows that among other factors, there is a significant correlation between multi-antibiotic resistance calculated indices from aquaculture-related bacteria and temperature (Table 2). In addition, we tested the association between low- and middle-income countries and MAR. As such, we have found that countries such as Vietnam,

India, Pakistan and Bangladesh simultaneously displayed the highest levels of clinical (human) and aquaculture MAR and were exposed to the highest climatic vulnerability and temperature, suggesting LMICs were regions most at risk for the combined action of global warming and AMR occurrence.

To study the effect of global warming on aquaculture, we focused on studying how mortalities associated with bacterial diseases are related to temperature. We agree with referee #1 that antibiotic use will not directly increase as a cause of higher mortalities, because the decision to treat with antibiotics is generally made prior to observing high mortalities. We however argue, as referee #2 stated, that antimicrobial use will have to increase in response to fish health challenges (which might not only include higher mortalities but also an increase in disease frequency), we have assessed this point in lines 220-223. We also agree that showing a clear increase in the frequency of aquaculture outbreaks in response to increasing temperatures is of the utmost importance. However, as mentioned by referee #1, there is a clear lack of published data to directly link a global increase in bacterial disease frequency to temperature. When we conceived the article, we tried to explore this subject consistently but failed to establish the association referee #1 proposed because of a lack of data in the literature.

Therefore, we acknowledge that this study shows an association between temperature and antibiotic resistance, but not a causality.

After rereading the article it seems as though the authors have not done a multiple regression analysis to assess or correct for confounding between temperature and LMIC (data in Table 2). Before the authors say that temperature is significantly associated with antibiotic resistance they should control for LMIC (these countries are often in warm locations and have poorer sanitary services, higher population densities, etc...) so some of the MAR attributed to temperature (which I am assuming is a proxy for higher bacterial replication and increased number of outbreaks) may have been from other factors that are often found in warm countries. I agree with the authors in principle that warm temperature will likely lead to increased outbreaks for most fish bacterial diseases and therefore an increase in the use of antibiotics, which will consequently increase the risk of AMR, but they need to control for confounding (e.g. other drivers of AMR that are also associated with temperature) before they can make the statement. I also think they should test for effect modification (i.e. warmer temperatures may have stronger association with MAR for LMIC than non LMIC) as it makes biological sense that it would occur.

At this stage, we can only speculate on the mechanisms involved, supporting referee #1 comment: “the underpinning mechanisms are likely to be multifactorial and could actually be related to the increase use of antimicrobials at warmer temperatures due to more bacterial infections in the population”. To address this concern, we have now modified the text throughout (lines 204-226) to unambiguously describe an association rather than a causal link, which was not the scope of our study. We have also strengthened the link between bacterial disease frequencies and temperature throughout the text, with the inclusion of additional references (FAO 2018, Bondad-Reantaso & Melba 2016), which report increases in bacterial disease in aquaculture farms as a result of increased temperature. These are expert reports and occasional field evidence, but there is a lack of systematic reporting of bacterial mortalities in aquacultures worldwide. Finally, since our paper submission, a study has been published in Science (Van Boeckel *et al.* 2019) showing similar trends between LMIC countries and MAR in farmed terrestrial animals (chickens, pigs). This reference has now been added in

the discussion to strengthen our findings.

We hope that this general response will satisfy the general comments of referee #1, but we have provided specific text modifications aimed at addressing all specific comments in the point by point responses below.

References:

- Bondad-Reantaso & Melba. Acute hepatopancreatic necrosis disease (AHPND) of penaeid shrimps: Global perspective. SEAFDEC <http://hdl.handle.net/10862/3084> (2016).
- FAO. *Impacts on climate change on fisheries and aquaculture – Synthesis on current knowledge, adaptation and mitigation options*. Rome. (2018).
- Van Boeckel, T.P., et al. Global trends in antimicrobial resistance in animals in low- and middle- income countries. *Science* 365, 1266 (2019).

Dear Editor,

Many thanks for the received comments and edits. We have redrafted the manuscript accordingly. Please find below our answer to reviewer #1

Best wishes,

Miriam Reverter & Rodolphe Gozlan

Reviewer #1

After rereading the article it seems as though the authors have not done a multiple regression analysis to assess or correct for confounding between temperature and LMIC (data in Table 2). Before the authors say that temperature is significantly associated with antibiotic resistance they should control for LMIC (these countries are often in warm locations and have poorer sanitary services, higher population densities, etc...) so some of the MAR attributed to temperature (which I am assuming is a proxy for higher bacterial replication and increased number of outbreaks) may have been from other factors that are often found in warm countries. I agree with the authors in principle that warm temperature will likely lead to increased outbreaks for most fish bacterial diseases and therefore an increase in the use of antibiotics, which will consequently increase the risk of AMR, but they need to control for confounding (e.g. other drivers of AMR that are also associated with temperature) before they can make the statement. I also think they should test for effect modification (i.e. warmer temperatures may have stronger association with MAR for LMIC than non LMIC) as it makes biological sense that it would occur.

We are pleased to see that all referees have agreed that warm temperatures will likely increase outbreaks for most fish bacterial diseases and that therefore an increase in the use of antibiotics will consequently increase the risk of AMR as this represents the core of our findings. However, in order to address reviewer #1 additional request, we have now studied 1) pair-wise correlations between all studied simple variables and displayed them in a correlation network to investigate the underlying relationship between them and 2) performed a multiple regression model with the variables correlated to aquaculture-derived MAR and calculated their relative importance as well as their interaction using the Lindeman, Merenda and Gold (LMG) metrics from the relaimpo package. We have also added population density as variables, following reviewer #1 suggestion. Composite variables combining several simple variables (e.g. in indices or scores such as HDI, EPI, CVI) were excluded from these analyses, because they often included redundant variables (e.g. GDP) that make interpretation more difficult. Secondly, since explanatory variables were highly correlated, we decided to use LMG to partition the total variance explained by the model, but we did not focus on interpreting the regressor coefficients and their significance (p-value), since in high multicollinearity scenarios, such as here, these estimates are highly sensitive to different variable combinations.

We understand the point raised by the reviewer and we agree that antibiotic resistances are clearly multi-factorial, involving socioeconomic, environmental and health-related factors, which are inherently interconnected. We made that point clear in our manuscript (lines 66-69 and 204-217). For example, we found that the clinical MAR is the variable explaining the highest variance of aquaculture-related MAR. Whereas horizontal gene transfer between human and animal bacteria is well known and is probably the main driver behind the strong association between aquaculture and clinical MAR, clinical MAR was also highly correlated to other socioeconomic variables, including the quantity of antibiotics used, the temperature and the percentage of undernourishment (correlation network, Figure 1). This is also underpinned by the recently published editorial of *The Lancet*¹, which states that 'the next steps must include

joint action by the human and animal health sectors'. We have discussed such complexity in the discussion (lines 204-212): "We found a strong correlation between MAR indices from aquaculture and MAR indices from human clinical bacteria, suggesting that different activities (human, livestock and aquaculture antimicrobial consumption) contribute to a common pool of AMR. Higher AMR levels in LMICs can be linked to factors such as poorer sanitation systems or antibiotic misuse and highlight the need to establish regulations, controls and information systems in those countries^{36,42-43}."

In order to explore these co-founding factors highlighted by the reviewer, we calculated the relative importance of the correlated variables to the aquaculture-related MAR (GDP per capita, temperature and clinical MAR) as well as the interaction between temperature and GDP per capita (following the selection of the model explaining the highest variance). These results highlight that the interaction between GDP per capita and temperature explains 9.7% of the variance of aquaculture-derived MAR, but they also show that temperature alone explains 9.1% of this variance (Figure 2), whereas GDP alone accounts for only 3%. This supports a direct involvement of temperature in aquaculture-related antibiotic resistance, as was cautiously discussed in the former version of the manuscript (lines 200-204). "In addition, we found that higher AMR levels of aquaculture-related bacteria were correlated with warmer temperatures, an association that has recently been observed amongst human clinical bacteria in the United States³⁵. Although drivers behind this association are still unclear and are likely multi-factorial, these could include higher use of antimicrobials linked to increases in disease frequency at higher temperatures⁴⁴."

For this reason, we consider , that although interesting, these analyses do not bring essential information to the main body of the article and we would prefer (unless the reviewer or editorial office thinks otherwise) to include the LMG results (figure 2) in the supplementary material in order to keep our message as straight as possible: the challenges of aquaculture in a scenario of global warming and increasing threats of antibiotic resistance.

1 Editorial Lancet (2020) The antimicrobial crisis: enough advocacy, more action. Vol 395 January 25, 2020.

Figure 1. Pearson correlation network between all the simple studied variables. Significant correlations (P-value < 0.05) are displayed with solid lines, whereas correlations ($r > 0.30$) nearing statistical significance ($0.10 > P > 0.05$) are shown in dashed lines. Edge weight is

proportional to the correlation coefficient (r), with line width increasing with higher correlation values.

Relative importances for MAR.aquaculture with 95% bootstrap confidence intervals

Figure 2. Relative importance of regressors from the multiple regression model (aquaculture MAR ~ Temperature + GDP capita + Clinical MAR + Temperature*GDP capita) with 95% bootstrap confidence intervals (1000 permutations) assessed by the Lindeman Merenda and Gold (LMG) metrics of the relaimpo R package.